# Can Rice Growth Substrate Substitute Rapeseed Growth Substrate in Rapeseed Blanket Seedling Technology? Lesson from Reactive Oxygen Species Production and Scavenging Analysis

**DOI:** 10.3390/antiox13081022

**Published:** 2024-08-22

**Authors:** Kaige Yi, Yun Ren, Hui Zhang, Baogang Lin, Pengfei Hao, Shuijin Hua

**Affiliations:** 1Institute of Crop and Nuclear Technology Utilization, Zhejiang Academy of Agricultural Sciences, Hangzhou 310021, China; kaige_y0113@163.com (K.Y.); linbg@mail.zaas.ac.cn (B.L.); 11816004@zju.edu.cn (P.H.); 2Institute of Crop, Huzhou Academy of Agricultural Sciences, Huzhou 313000, China; yunhuaren@163.com; 3Zhejiang Agri-Tech Extension and Service Center, Hangzhou 310020, China; zhanghui881007@126.com

**Keywords:** ascorbic acid, *Brassica napus* L., crop growth substrate, enzyme, glutathione, reactive oxygen species, seedling quality

## Abstract

Rapeseed (*Brassica napus* L.) seedlings suffering from inappropriate growth substrate stress will present poor seedling quality. However, the regulatory mechanism for the production and scavenging of reactive oxygen species (ROS) caused by this type of stress remains unclear. In the current study, a split plot experiment design was implemented with two crop growth substrates—a rice growth substrate (RIS) and rapeseed growth substrate (RAS)—as the main plot and two genotypes—a hybrid and an open-pollinated variety (Zheyouza 1510 and Zheyou 51, respectively)—as the sub-plot. The seedling quality was assessed, and the ROS production/scavenging capacity was evaluated. Enzymatic and non-enzymatic systems, including ascorbic acid and glutathione metabolism, and RNA-seq data were analyzed under the two growth substrate treatments. The results revealed that rapeseed seedling quality decreased under RIS, with the plant height, maximum leaf length and width, and aboveground dry matter being reduced by 187.7%, 64.6%, 73.2%, and 63.8% on average, respectively, as compared to RAS. The main type of ROS accumulated in rapeseed plants was hydrogen peroxide, which was 47.8% and 14.1% higher under RIS than under RAS in the two genotypes, respectively. The scavenging of hydrogen peroxide in Zheyouza 1510 was the result of a combination of enzymatic systems, with significantly higher peroxidase (POD) and catalase (CAT) activity as well as glutathione metabolism, with significantly higher reduced glutathione (GSH) content, under RAS, while higher oxidized glutathione (GSSH) was observed under RIS. However, the scavenging of hydrogen peroxide in Zheyou 51 was the result of a combination of elevated oxidized ascorbic acid (DHA) under RIS and higher GSH content under RAS. The identified gene expression levels were in accordance with the observed enzyme expression levels. The results suggest that the cost of substituting RAS with RIS is a reduction in rapeseed seedling quality contributing to excessive ROS production and a reduction in ROS scavenging capacity.

## 1. Introduction

Rapeseed is an important oil crop distributed mainly in Canada, the European Union, China, India, and Austria. China has produced nearly 1.55 billion kilograms annually in recent years [1]. However, the heavy consumption of edible vegetable oil due to its large population has resulted in a high demand for vegetable oil, and it has been estimated that the self-sufficiency rate of vegetable oil in China is only about 30%. Therefore, seeking potential pathways to increase the seed yield of rapeseed for the alleviation of the deficient supply of vegetable oil is necessary.

There are two main methods for increasing rapeseed seed yield. One is genetic manipulation, which mainly focuses on the improvement in yield-related traits. For example, Shah et al. (2018) used RNA-Seq technology at different developmental stages and temperature regimes in a semi-winter canola cultivar to identify key genes related to yield traits. They identified 36 genes associated with flowering and yield, which are useful for further genetic manipulation [2]. Liu et al. (2024) identified a gene, *BnaA4.BOR2*, which is critical for the transportation of boron from roots to shoots, with its loss of function leading to enhanced sensitivity to low boron supply in soil [3]. Ye et al. (2024) identified a causal gene, *BnaA9.CYP78A9*, which has a heterotic effect, resulting in partial dominance of auxin expression and, eventually, heterosis of the cell number [4]. The mentioned examples were the tip of the iceberg in the progress of genetic manipulation for improvement in rapeseed yield. Another way to improve yield is through agronomic practices, including adjustment of tillage methods, improvement in fertilizer use efficiency, arrangement of irrigation, and control of diseases and insects [5,6,7,8]. In rapeseed production, two planting methods are used: direct seeding and transplanting. Transplanting is an effective planting method to gain high rapeseed yield; however, the big problem of this method is in need of manual operations. Thus, the application of this conventional method has been sharply decreasing in recent decades due to the insufficient labor force. In order to retain the advantages of transplanting rapeseed, the blanket seedling transplanting technology has been successfully applied in China, which solves the bottleneck of transplanting mechanization [9]. Unlike conventional transplanting, with the plants individually being transplanted, the high density of seedlings in a seed tray and interlocking among plant roots are the main characteristics of the blanket seedling transplanting method. Therefore, ensuring the high quality of the rapeseed seedlings obtained using blanket seedling transplanting technology is quite important to guarantee high rapeseed yield. There are many factors affecting seedling quality during the cultivation of rapeseed seedlings, including the growth substrate [10,11,12,13]. As blanket seedling transplanting technology is used not only for rapeseed, but also in rice production [14,15], farmers have considered the substitution of a rice seedling growth substrate for a rapeseed growth substrate under the rice–rapeseed rotation mode in China, which can save both costs and time. However, due to the difference in components between the two growth substrates, the effect of a rice seedling growth substrate on rapeseed growth is hypothetically different. The subsequent question arises: how do different seedling growth substrates regulate rapeseed seedling quality?

Rapeseed is easily affected by many abiotic stresses, such as drought, waterlogging, salt stress, low and high temperature stress, and nutrient deficiency at the seedling stage [16,17,18]. When plants are subjected to adverse abiotic stresses, positive responses to those stresses could be rapidly recalled, benefiting from the strong self-plasticity of the plant [19]. In such a cellularly programmed procedure, the antioxidant system—including enzymatic and non-enzymatic systems—is a useful defense system to maintain crop health under different abiotic stresses. In this system, many reactive oxygen species (ROS)—mainly including superoxide anions, hydrogen peroxide, and hydroxyl radicals—can be scavenged by corresponding enzymes, including superoxide dismutase (SOD), peroxidase (POD), and catalase (CAT), as well as through other non-enzymatic pathways, such as ascorbic acid (ASA) and glutathione (GSH) metabolism [20,21,22,23]. Many investigations have been carried out to describe ROS production and the regulation of ROS scavenging in different crops under different stresses. For example, maize root growth was affected by the ROS homeostasis through the inverse regulation of *ZmUPB1* and *ZmPRX12* transcription in the cells of the root apex transition zone in response to nitrogen depletion or nitrate supply [24]. The effects of climate change, such as abrupt temperature reductions, are important factors causing ROS production in plants, as well as affecting metabolism at the seedling stage. In wheat, an abrupt temperature reduction resulted in a considerable accumulation of ROS and leaf cell death in a temperature-sensitive genotype [25]. Although similar examples can be listed, very few researchers have paid attention to the impacts on crop development via ROS metabolism caused by different crop growth substrates. Furthermore, the influences should be considered comprehensively, as the stressor is not a simple and single factor. However, investigations in this context are scarce, especially considering that rapeseed blanket seedling transplanting technology has only been recently developed in China.

In the current study, we used two substrates (those for rice and rapeseed seedling cultivation) and two types of a rapeseed cultivar (an open pollinator and a hybrid), in order to compare the quality of the resulting seedlings. Further, a systematic analysis of ROS metabolism was conducted in the rapeseed seedlings under the two growth substrates. Finally, the regulatory mechanism was also assessed through an RNA-sequencing analysis. The goals of this investigation were 1. to explore the effects of rice and rapeseed growth substrates on rapeseed seedling quality; 2. to identify the variations in response to the two growth substrates in the aspect of ROS metabolism; and 3. to uncover the molecular regulatory mechanism of ROS metabolism caused by the two growth substrates. The results are expected to provide evidence regarding the possibility of substituting a rapeseed growth substrate with a rice growth substrate.

## 2. Materials and Methods

### 2.1. Plant Material and Experiment Design

The experiment was conducted from 15 October to 15 November in 2023 at the experimental station of Zhejiang Academy of Agricultural Sciences, Hangzhou, China. The experimental design was a split plot experimental design. Two growth substrates—specific for rice (RIS) and rapeseed (RAS)—served as the main plot; meanwhile, two cultivars—namely, open-pollinated (Zheyou 51) and hybrid (Zheyouza 1510) varieties—were employed as a sub-plot. Each experimental treatment was replicated 3 times with 9 trays. The RIS and RAS were purchased from Jiangsu Peilei Substrate Technology Company Limited (Zhenjiang, Jiangsu, China) and Jiangsu Xingnong Substrate Technology Company Limited (Zhenjiang, Jiangsu, China). The main components of RIS were humus, peat, plant ash, and sand, while the components of RAS were a fermented mixture of mushroom residue, biogas residue, cassava residue, coconut husk, peat, vermiculite, perlite, and plant ash. The nutrients in the two substrates are listed below: total N, P, and K were 21.5, 9.2, and 5.6 g kg^−1^ in RIS and 43.1, 23.0, and 12.6 g kg^−1^ in RAS. The pH value was 5.80 and 6.16 in RIS and RAS, respectively, and the organic matter was 4.78% and 7.83% in RIS and RAS, respectively. Two rapeseed varieties were chosen with the aim of comparing whether the hybrid (Zheyouza 1510) has significantly stronger heterosis than the open-pollinated variety (Zheyou 51). The RIS and RAS were filled in the tray to 3.0 cm in depth. Then, the seeds were seeded using an automatic sowing machine (2BSL-320, Jiangsu Yunma Machinery Manufacturing Company Limited, Yancheng, Jiangsu, China), at 1000 seeds per tray. The seeds were covered with a thin layer of substrate. Each treatment was composed of 10 trays. The trays were stacked with sowed seeds for 36–48 h. Then, the trays were arranged in a plain place and it was checked whether the soil presented water deficiency. The trays were then covered with a layer of white nonwovens with a weight of 30–50 g cm^−1^. The nonwovens were uncovered after 36–48 h. After this step, topdressing was required. The amount of urea was 1 g tray^−1^ at the one-leaf and two-leaf stages, respectively. For topdressing, the urea was dissolved into water as a foliar fertilizer. The application timing should be either in the morning or evening, but not at noon.

### 2.2. Sampling

The seedlings were sampled at 30 days of age (Figure 1). The seedlings were uniformly taken, with the mixture of 5 plants per tray. The plants at the border of the tray were excluded, in order to avoid any marginal effects. The plants were rinsed carefully with distilled water until the soil on their roots had been clearly removed. The plants were divided into aboveground and underground parts for dry matter determination. The third leaf was taken for ROS production and enzyme activity measurement. Another piece of a leaf sample was put into liquid nitrogen immediately and then transferred to a −80 °C refrigerator until the RNA-seq analysis.

### 2.3. Seedling Quality Assessment

Seedling quality parameters were assessed under the two crop growth substrates and genotypes, including plant height, maximum leaf length and width, and aboveground and underground biomasses. The plant height and the maximum leaf length and width were recorded using a ruler. The plants were divided into aboveground and underground parts and the samples were killed at 90 °C for 30 min, then dried at 70 °C until a constant weight had been reached.

### 2.4. ROS Production and Enzymatic Activity Analysis

#### 2.4.1. ROS Production and Scavenging Capacity Analysis

The production of ROS, including hydrogen peroxide and superoxide anions (SOAs), was measured in this study. The methods for measurement were conducted according to an assay kit from Suzhou Comin Biotechnology Company, Limited (Suzhou, Jiangsu, China). The hydrogen peroxide and superoxide anion content analyses used the titanium sulfate colorimetric method and hydroxylamine oxidation method, respectively [26]. The hydroxyl free radical scavenging (HFRS) capacity analysis used the ortho phenanthrene method [27]. The superoxide anion scavenging capacity analysis used the sulfonamide colorimetric method [28], and total antioxidant capacity was assessed using the diammonium 2,2′-azino-bis (3-ethylbenzothiazoline-6-sulfonate) (ABTS) method [29].

#### 2.4.2. Enzymatic Analysis for ROS Scavenging

The activities of enzymes including superoxide dismutase (SOD), catalase (CAT), and peroxidase (POD) were measured according to the instructions of an assay kit from Suzhou Comin Biotechnology Company, Limited (Suzhou, Jiangsu, China). The SOD, CAT, and POD activity measurements used the WST-8 method, ammonium molybdate colorimetric method, and guaiacol colorimetric method, respectively [30].

#### 2.4.3. Ascorbic Acid Pathway Analysis

The ascorbic acid pathway analysis included the contents of reduced ascorbic acid (AsA), oxidized ascorbic acid (DHA), and their metabolic enzymes. The AsA and DHA contents were measured using the fast blue B salt colorimetric method and dithiothreitol (DTT) reduction method, respectively. The activities of various enzymes—including ascorbate peroxide (APX), dehydroascorbate reductase (DHAR), monodehydroascorbate reductase (MDHAR), ascorbate oxidase (AAO), and L-galactono-1,4-lactone dehydrogenase (Gal LDH)—were measured according to the instructions of an assay kit from Suzhou Comin Biotechnology Company, Limited (Suzhou, Jiangsu, China), and following the method described in [31].

#### 2.4.4. Glutathione Pathway Analysis

The reduced (GSH) and oxidized (GSSG) glutathione contents and their associated metabolic enzymes, including glutathione peroxidase (GPX), glutathione reductase (GR), glutathione S-transferase (GST), thioredoxin peroxidase (TPX), and glutamate cysteine ligase (GCL), were assessed. The GSH and GSSG contents and the activities of the associated enzymes were measured according to the instructions of an assay kit from Suzhou Comin Biotechnology Company, Limited (Suzhou, Jiangsu, China), and following the method described in [32].

#### 2.4.5. Other Antioxidants

The total sulfhydryl group, non-protein sulfhydryl group, and protein carbonyl contents were measured according to the instructions of an assay kit from Suzhou Comin Biotechnology Company, Limited (Suzhou, Jiangsu, China). The total sulfhydryl group and non-protein sulfhydryl group contents were determined using the 5,5-dithiobis-2-nitrobenzoic acid (DTNB) colorimetric method [33]. The protein carbonyl contents were determined using the 2,4-dinitrophenylhydrazine colorimetric method [34].

### 2.5. RNA-Seq Analysis

Leaf samples of Zheyouza 1510 under RIS and RAS treatments were used for total RNA extraction, according to the instructions of a polysaccharide and polyphenol total RNA isolation kit (Bioteke, Beijing, China). The libraries were constructed according to the instructions and then sequenced on an Illumina Nova-seq 6000 system (Illumina Inc. (San Diego, CA, USA)). HISAT2 was used to align all the clean reads against the reference genome for *Brassica napus* [35]. The expression levels of transcripts were expressed according to the fragments per kilobase of exon model per million mapped fragments (FPKM) values. The differentially expressed genes (DEGs) were defined as those with |log2FoldChange| > 1 and *p*-value ≤ 0.05, according to the DEseq2 methods and R program (R program, Version 4.4.1) [36]. The genes encoding enzymes associated with ROS scavenging were the focus of this analysis.

### 2.6. Quantitative Real-Time Polymerase Chain Reaction (qRT-PCR) Analysis

In order to validate the RNA-seq data, the correlations between qRT-PCR and RNA-seq data were obtained for ROS production and scavenging. For qRT-PCR, the leaf samples in Zheyouza 1510 under the two crop growth substrates were used for RNA extraction, according to the instructions of the QIAGE kit (4016050, Hangzhou Simgen Biotechnology Co., Ltd., Hangzhou, China). The total RNA concentration was then measured using a Thermo ultramicro spectrophotometer (NanoDro 2000C, Shanghai Bajiu Industrial Co., Ltd., Shanghai, China). cDNA was transcribed according to the instructions, using PrimerScriptTM 1st Strand cDNA synthesis (Perfect Real Time) (6110A, Takara, TaKaRa Biotechnology (Dalian) Co., Ltd., Kusatsu, Japan). The quantification of gene expression was conducted on a Roche LightCyle 480 fluorescence quantitative PCR instrument (Roche, Basel, Switzerland) with the TB Green Primix Ex Taq TM II (Tli RNaseH Plus) and ROX plus enzyme kit (Takara, Kyoto, Japan). Rapeseed *ACTIN* was used as the reference gene, and the sequences of the primers used in this study are listed in Appendix A. There were 2, 6, 1, 2, 5, 4, and 4 genes encoding GPX, GST, GR, TPX, CAT, POD, and SOD selected in this study, respectively (Appendix A).

### 2.7. Statistics

The data are expressed as mean values (n = 3). The SPSS statistical software (version 26.0, Chicago, IL, USA) was employed for an analysis of variance (ANOVA). Duncan’s multiple range test (DMRT) was used to evaluate significant treatment effects at the significance level of *p* ≤ 0.05. The correlations between leaf agronomic traits and ROS contents were obtained using Pearson’s correlation analysis. The significant differences in the gene expression amount were analyzed (*p* < 0.05). The transcript levels were calculated using the 2^−ΔΔCt^ method [37]. The transcriptomic data presented in this study can be found in online repositories using the following accession number: PRJNA1140847 (National Center for Biotechnology Information, NCBI, Bethesda, MD, USA).

## 3. Results

### 3.1. Seedling Quality and ROS Production under Different Crop Growth Substrates and Genotypes

The 30-day-old rapeseed seedlings under rice and rapeseed growth substrates showed significantly different morphologies (Figure 1). For both genotypes, seedlings under RAS had significantly better agronomic performance than those under RIS (Table 1). The plant height, maximum leaf length and width, and aboveground dry matter of Zheyouza 1510 seedlings under RIS were decreased by 119.7%, 35.4%, 38.6%, and 19.2%, respectively, when compared to RAS. However, there was no significant difference in underground dry matter between the two growth substrates for Zheyouza 1510. For Zheyou 51, the above-mentioned agronomic traits decreased by 71.4%, 42.2%, 45.3%, 51.9%, and 61.5%, respectively, under RIS as compared to RAS treatment. All agronomic traits, except for underground dry matter, were significantly better for Zheyou 51 than Zheyouza 1510 under the RAS treatment; in particular, the plant height, maximum leaf length and width, and aboveground and underground dry matter in Zheyou 51 were 1.74-, 1.09-, 1.19-, and 1.52-fold better than those for Zheyouza 1510. However, under RIS treatment, most agronomic traits (except for underground dry matter) presented no significant difference between the two genotypes (Table 1).

In order to explore the reasons for the differences in agronomic performance between the rapeseed seedlings, the production of ROS was evaluated. The results showed that the hydrogen peroxide content under RIS was significantly higher than that under RAS in both genotypes, showing increments of 47.7% and 14.1%, respectively. For the variation of genotypes, Zheyouza 1510 produced significantly less hydrogen peroxide under RAS, which was decreased by 25.3% when compared to Zheyou 51. There was no significant difference in hydrogen peroxide content between the two genotypes under RIS treatment (Figure 2A).

Unlike hydrogen peroxide content, the SOA content in Zheyouza 1510 was significantly higher than that in Zheyou 51 under the two growth substrates, with increments of 2.2% and 2.5% under RIS as compared to RAS, respectively. Regarding the effect of the crop growth substrate on SOA production in rapeseed leaves, no significant difference was found between the two genotypes (Figure 2B).

In order to further reveal the relationships between seedling quality and ROS production, a Pearson correlation analysis was conducted. The results indicated that both hydrogen peroxide and superoxidase anion contents were negatively correlated with rapeseed agronomic traits. Furthermore, the hydrogen peroxide content was significantly negatively associated with maximum leaf length and width (Figure 2C).

### 3.2. ROS Scavenging Capacity under Different Crop Growth Substrates and Genotypes

It is important to scavenge ROS in a timely manner to maintain plant health. The results indicated that both genotypes had stronger hydroxyl radical scavenging (HFRS) capacity under RAS than under RIS, which was increased by 4.9 times for both genotypes. The HFRS capacity in Zheyou 51 was significantly higher than that in Zheyouza 1510 under the two crop growth substrates, which was increased by 53.0% for both substrates (Figure 3A).

For superoxide anion scavenging (SOAS) capacity, it was observed that Zheyou had a significantly higher capacity under the two growth substrates, when compared to Zheyouza 1510, showing increments of 54.9% and 87.3%, respectively. Significantly higher SOAS capacities under RAS were found, when compared to RIS treatment, in both genotypes, showing increments of 19.9% and 29.2%, respectively (Figure 3B).

For total antioxidant capacity (T-AOC), it was found that the capacity of Zheyouza 1510 was significantly higher than that of Zheyou 51 under the two growth substrates, with increments of 18.4% and 27.3%, respectively. The T-AOC capacity under RIS was 15.8% and 9.5% lower than that under RAS treatment in Zheyouza 1510 and Zheyou 51, respectively (Figure 3C).

### 3.3. Enzymatic Activities Analysis on ROS Scavenging Capacity under Different Crop Growth Substrates and Genotypes

Superoxide dismutase (SOD), peroxidase (POD), and catalase (CAT) are three main enzymes for the elimination of ROS during plant development under environmental stresses. It was observed that the activity of SOD in Zheyou 51 was significantly higher than that in Zheyouza 1510 under the two growth substrates, which showed increments of 15.3% and 37.5%, respectively. Regarding the influence of crop growth substrates on the activity of SOD, there was no significant difference in Zheyouza 1510. However, the SOD activity under RAS was significantly higher than that under RIS in Zheyou 51, which showed an increment of 28.2% (Figure 4A).

For POD activity, there was no significant difference between the two crop growth substrate treatments in Zheyou 51; however, the POD activity under RAS was significantly higher than that under RIS in Zheyouza 1510, which showed an increment of 47.2%. For genotypic variation, no significant difference in POD activity was found between the two genotypes under the RAS treatment. However, the POD activity in Zheyou 51 was significantly higher than that in Zheyouza 1510, which showed an increment of 30.1% (Figure 4B).

For CAT activity, it was found that the activity under RAS was significantly higher than that under RIS in Zheyouza 1510, which showed an increment of 68.5%. There was no significant difference between the two crop growth substrates for Zheyou 51. For genotypic variations, the CAT activity in Zheyou 51 was 18.9% higher than that in Zheyouza 1510 under RIS. However, the reverse trend was found under RAS treatment, revealing a decrement of 21.7% in Zheyou 51 as compared to Zheyouza 1510 (Figure 4C).

### 3.4. Ascorbic Acid Metabolism under Different Crop Growth Substrates and Genotypes

Ascorbic acid metabolism is one of the important pathways for coping with excessive ROS production in plant cells. The reduced ascorbic acid (AsA) content in Zheyouza 1510 was significantly higher than that in Zheyou 51, with increments of 11.9% and 18.4% under RIS and RAS treatments, respectively. There was no significant difference between rice and rapeseed growth substrate treatments in Zheyouza 1510, while a significantly higher content (5.9% increment) under RIS as compared to RAS was observed for Zheyou 51 (Figure 5A).

Regarding oxidized ascorbic acid (DHA) content, there was no significant difference between the two crop growth substrates for Zheyouza 1510. However, a significant difference in DHA content between the two substrates was observed in Zheyou 51, with a 39.4% increment under the RIS treatment as compared to RAS. For genotypic variation, the DHA content under RIS in Zheyou 51 was significantly higher than that in Zheyou 1510, which showed an increment of 67.6%. The DHA content under RAS did not significantly differ between the two genotypes (Figure 5B).

For dehydroascorbate reductase (DHAR), the activity in Zheyouza 1510 under RIS treatment was significantly higher than that under RAS treatment, which showed an increment of 29.8%. However, there was no significant difference observed between the two growth substrates for Zheyou 51. Regarding the genotypic difference, there was no significant difference under RIS between Zheyouza 1510 and Zheyou 51. However, under RAS treatment, the DHAR activity in Zheyou 51 was significantly higher than that in Zheyouza 1510, which showed an increment of 40.3% (Figure 5C).

Considering the activities of monodehydroascorbate reductase (MDHAR) and ascorbic acid oxidase (AAO), a similar trend was observed. Significantly higher MDHAR and AAO activities under RIS were found both in Zheyouza 1510 and Zheyou 51, as compared to RAS, which showed the increments of 116.2% and 118.2% (MDHAR) and 100.9% and 138.7% (AAO), respectively. There was no genotypic variation under the two crop growth substrates (Figure 5D,E).

For ascorbate peroxidase (APX) activity, there was no significant difference between the two crop substrates in both Zheyouza 1510 and Zheyou 51. However, significant genotypic variations were found. The APX activity in Zheyou 51 was significantly higher than that in Zheyouza 1510 under RIS and RAS treatments, which showed increments of 40.2% and 42.9%, respectively (Figure 5F).

For L-galactose-1,4-lactone dehydrogenase (Gal LDH) activity, no significant differences were observed between the two crop growth substrates for both Zheyouza 1510 and Zheyou 51. Regarding genotypic variation, the Gal LDH activities under the two crop growth substrates in Zheyouza 1510 were significantly lower than those in Zheyou 50, which showed decrements of 37.6% and 34.1%, respectively (Figure 5G).

### 3.5. Glutathione Metabolism for ROS Scavenging under Different Crop Growth Substrates and Genotypes

Substantial reduced glutathione (GSH) accumulated under RAS in both Zheyouza 1510 and Zheyou 51, which showed increments of 257.8% and 108.6% when compared to RIS, respectively. For genotypic variation, the GSH content in Zheyouza 1510 was 2.0 times higher than that in Zheyou 51 under RAS. However, there was no significant difference in GSH content between the two genotypes under RIS (Figure 6A).

For the oxidized glutathione (GSSG) content, there was no significant difference between the two crop growth substrates in Zheyou 51. However, the content under RIS was significantly higher than that under RAS in Zheyouza 1510, which showed an increment of 11.1%. For genotypic variation, the GSSG contents under the two crop growth substrates in Zheyou 51 were significantly higher than those in Zheyouza 1510, which showed increments of 9.1% and 13.5%, respectively (Figure 6B).

The activity of glutathione peroxidase (GPX) in Zheyouza 1510 presented no significant difference between the two crop growth substrates. However, the GPX activity under RAS was significantly higher than that under RIS in Zheyou 51, which showed an increment of 44.3%. For the genotypic variation, it was observed that the GPX activities in Zheyou 51 were significantly higher than those in Zheyouza 1510 under the two crop growth substrates, which showed increments of 36.7% and 80.1%, respectively (Figure 6C).

The glutathione reductase (GR) activity under RIS was significantly higher than that under RAS in both Zheyouza 1510 and Zheyou 51, which showed increments of 8.7% and 17.4%, respectively. For genotypic variation, significantly higher GR activity in Zheyou 51 was observed under both crop growth substrates when compared to Zheyouza 1510, with increments of 29.2% and 19.6%, respectively (Figure 6D).

Regarding glutathione-S-transferase (GST) activity, no significant differences were detected, both between the two crop growth substrates and the genotypes (Figure 6E).

For the thioredoxin peroxidase (TPX) activity, significantly higher activity under RAS was observed in the two genotypes, when compared to RIS, showing increments of 60.6% and 149.9%, respectively. For genotypic variation, the TPX activity under RIS did not significantly differ between the two genotypes. However, the TPX activity in Zheyou 51 was significantly higher than that in Zheyouza 1510 under RAS, which showed an increment of 53.1% (Figure 6F).

Glutamate cysteine ligase (GCL) activity was not detected under RIS in both genotypes. Significantly higher GCL activity was found under RAS in Zheyou 51, which showed an increment of 231.9% when compared to Zheyouza 1510 (Figure 6G).

The total sulfhydryl (TSH) content under RAS was significantly higher than that under RIS in Zheyouza 1510, which showed an increment of 84.5%. Meanwhile, there was no significant difference between the two crop growth substrates in Zheyou 51. The TSH content under RIS was significantly higher in Zheyou 51 as compared to Zheyouza 1510, which showed an increment of 27.3%. A reverse trend was found under RAS, which had 59.3% higher TSH content in Zheyouza 1510 as compared to Zheyou 51 (Figure 6H).

### 3.6. RNA-Seq Analysis of the ROS Scavenging Capacity under Different Crop Growth Substrates and Genotypes

In order to further reveal the molecular ROS regulation mechanism under two crop growth substrates regarding rapeseed seedling quality, an RNA-seq analysis was performed in Zheyouza 1510 using RAS as a control. The volcano plot result showed that 1972 genes were significantly up-regulated and 2423 genes were significantly down-regulated (Figure 7A).

The KEGG results indicated that genes encoding metabolites involving the removal of ROS were enriched, such as phenylpropanoid biosynthesis and isoflavonoid biosynthesis (Figure 7B). In addition to those metabolic genes, genes involved in photosynthesis and plant hormones were highly enriched. To further identify genes in the non-enzymatic ROS scavenging system, specific genes in the RNA-seq data were chosen and their expression modes were analyzed. The results showed that 20 genes encoding GST were identified. Of those genes, 13 were down-regulated while 7 were up-regulated under RIS treatment. There were two, one, and five genes encoding GPX, GR, and AAO, respectively. The two identified GPX genes and one GR gene were up-regulated under RIS treatment. All identified AAO genes were new genes with unclear functions; furthermore, only one gene (*New-Gene_10590*) was up-regulated under RAS treatment. In the enzymatic system for ROS scavenging, 8, 12, and 8 genes encoding SOD, POD, and CAT were identified, respectively. For the genes encoding SOD, three and five genes were up- and down-regulated, respectively, under RAS treatment. For POD, twice the number of genes were up-regulated, compared to the down-regulated ones. For CAT, two genes—*BnaA07G0132000ZS* and *BnaA07G0132100ZS*—were up-regulated, while six genes were down-regulated under RAS treatment (Figure 7C). In order to validate the quality of the RNA-seq data, qRT-PCR was performed with randomly selected genes. The Pearson correlation between RNA-seq and qRT-PCR data was analyzed. The result indicated that the correlation coefficient reached 0.90, indicating the reliability of the RNA-seq data (Figure 7D).

## 4. Discussion

The differences in rapeseed agronomic trait performance observed in the current study were caused by the different crop growth substrates. However, the influences of the substrates on rapeseed seedling quality were a composite effect, resulting from the different compositions, nutrient contents, and compactness of the substrates. As no relevant investigations could be found in the existing literature, we considered a relatively new rapeseed transplanting technology and composite stress parameters. Therefore, a systematic analysis of the ROS production and scavenging performance was conducted in rapeseed seedlings to determine the reasons for the worse agronomic traits of rapeseed under RIS.

Rapeseed blanket seedling transplanting technology is a newly developed technology for solving the problem of mechanized transplanting in the field of rapeseed production [9]. Therefore, ensuring seedling quality is essential for a successful adoption of the technology. However, at present, farmers usually use a rice growth substrate for rapeseed seedling cultivation, as blanket seedling transplanting technology is also applied in rice production and it is believed that the application of RIS can reduce production costs [14,15]. However, poorer agronomic trait performance of rapeseed seedlings was observed under RIS treatment in this study. In particular, poor performance in terms of plant height, maximum leaf size, and dry matter was found (except for underground dry matter) in Zheyouza 1510 under RIS. The results support the viewpoint that the use of improper growth media can affect the growth and development of rapeseed plants [38,39]. Furthermore, significant differences in agronomic traits were observed between the two genotypes. Amazingly, the responses to RIS and RAS differed for the two genotypes. Under the normal growth substrate (RAS), the genotype Zheyouza 1510 did not show any heterosis. However, under RIS, no significant differences in agronomic traits were found between the two genotypes, except that the underground dry matter of Zheyouza 1510 was significantly higher than that of Zheyou 51. This result clearly showed that, for rapeseed plants under stress conditions, the heterosis potential of Zheyouza 1510 led to good results. Therefore, it is desirable to select a hybrid rapeseed variety when the plants may suffer from adverse stresses [40,41].

Hydrogen peroxide and superoxide anions are two important ROS [42,43]. In the current study, considerable hydrogen peroxide deposition under RIS was observed, compared to RAS, and this was the main type of ROS production in both genotypes. This result was in accordance with the negative correlation to agronomic performance in rapeseed plants under higher hydrogen peroxide production. However, the hybrid variety (Zheyouza 1510) was superior, in that it accumulated the least hydrogen peroxide content under RAS. However, this result did not lead to better agronomic performance in the rapeseed plants, indicating that seedling quality might be not only affected by the growth substrate [44,45]. Although the superoxide anion content did not significantly differ between crop growth substrate treatments, the genotypic variation was significant. Significantly higher SOA in Zheyouza 1510 might be expected to result in higher hydrogen peroxide content; however, the result was opposite. Furthermore, the superoxide anion scavenge capacity in Zheyouza 1510 was also lower than that in Zheyou 51. These results suggested that the ROS production and scavenging pathways differ between different genotypes [46,47].

Excessive ROS has adverse effects on plant growth and development, as revealed through the correlation analysis and previous investigations [48,49]. Timely removal of excessive ROS is important in plants. The SOD activity in Zheyouza 1510 was not high, while it was very high in Zheyou 51 under RAS, indicating that SOA can be catalyzed by this enzyme to produce hydrogen peroxide. Therefore, the reduction in both SOA and hydrogen peroxide is an important reason for the better agronomic performance of Zheyou 51 under the RAS treatment. In Zheyouza 1510, significantly higher POD and SOD activities were found under RAS, as compared to RIS. Those two higher enzymatic activities revealed the essential reason for the lower hydrogen peroxide content under RAS treatment. On the other hand, this result also suggested that the enzymatic pathway for ROS scavenging is important in Zheyouza 1510. However, the POD and CAT activities did not significantly differ in Zheyou 51 between the two growth substrate treatments. This result suggests that the antioxidant enzymatic system might not be the main pathway for scavenging of hydrogen peroxide in Zheyou 51. This provides further evidence for the different genotypic responses to various crop growth substrates from a ROS metabolism viewpoint [50,51].

In addition to the enzymatic catalysis system, the ascorbic acid pathway is another non-enzymatic system for the removal of excessive ROS. In the current study, there were no significant differences in reduced (AsA) and oxidase ascorbic acid (DHA) content in Zheyouza 1510 between the two growth substrates. This result indicates that the AsA metabolic pathway might not be the main pathway for coping with excessive ROS under the RIS treatment in Zheyouza 1510. The higher MDHAR and AAO activities are helpful to accumulate more AsA and DHA under the RIS treatment, achieving the balance between reduced and oxidized ascorbic acid. In crops, tradeoffs between different physiological metabolic pathways are common [52,53]. However, in Zheyou 51, both the AsA and DHA contents under RIS were significantly higher than those under RAS. Furthermore, the increase in DHA under the RIS treatment was obviously higher than that of AsA. The higher percentage of DHA might help to explain the higher hydrogen peroxide content under RIS in Zheyou 51. For the homeostasis of AsA and DHA content, similar MDHAR and AAO activities in Zheyou 51 were found under RIS treatment, as compared to Zheyouza 1510.

Glutathione metabolism is another important pathway for the removal of excessive ROS [54,55]. It was found that substantial GSH was deposited in Zheyouza 1510 rapeseed plants under the RAS treatment, which is good for clearing ROS. Furthermore, the significantly higher GSSG content indicated that the available GSH was reduced. The higher GSH content under RAS was due to the contribution of higher TPX and GCL activities. Significantly higher GSH content was also observed in Zheyou51 under RAS, but its content was lower than that in Zheyouza 1510. The higher GSH content under RAS in Zheyou 51 can be attributed to the higher activities of GPX, TPX, and GCL. Unlike Zheyou 51, a significantly higher TSH content was observed in Zheyouza 1510 under RAS treatment, indicating that higher TSH could be beneficial to mitigate the growth retardance due to excessive ROS production.

In order to further reveal the molecular regulation of ROS production and scavenging caused by the different crop growth substrates, RNA-seq and qRT-PCR analyses were performed. Under the RIS treatment, a number of significantly differentially negative expressed genes were activated, which could potentially explain the poor agronomic performance [56]. The further KEGG analysis showed that genes encoding antioxidants were identified in rapeseed plants under RAS, which was in accordance with their higher antioxidant capacity. It has been reported that a higher capacity of antioxidants, such as flavonoids and phenols, has positive effects on crop growth and development [57,58]. When identifying genes involved in the elimination of ROS, including enzymatic and non-enzymatic systems, a surprising result revealed that the significant differentially expressed genes between different growth substrates were not high in number. While a considerable number of genes encoding GST were identified, there were no significant differences in GST activity between the crop growth substrates. This might be due to the functions of these positive and negative genes offsetting each other [59,60]. The number of genes encoding POD with a positive mode under RAS was obviously higher than that with a negative mode, which is in agreement with the dynamics of POD activity observed under different crop growth substrate treatments. It also should be noted that, besides genes related to ROS metabolism, higher amounts of genes enriched in processes such as photosynthesis-antenna proteins and phytohormone signal transduction were also identified. Furthermore, the identified genes encoding antenna proteins were significantly down-regulated. The results suggest that the worse growth and development performance of rapeseed at the seedling stage might be the result of a combination of ROS metabolism and other pathways, such as photosynthetic metabolism, which should be further proved.

## 5. Conclusions

In the present study, rapeseed seedling quality was found to be seriously affected by RIS treatment, with reduced plant height, maximum leaf size, and dry matter accumulation being observed. The rapeseed leaf size was negatively associated with ROS production. Therefore, to ensure the high quality of rapeseed seedlings, a reduction in ROS deposition in plants is required. Both genotypes—a hybrid and an open-pollinated variety—revealed the main ROS type to be hydrogen peroxide under RIS, when compared to RAS. Consequently, the elimination of hydrogen peroxide in rapeseed plants is the most important issue when seeking to enhance the growth and development of rapeseed. In the ROS scavenging process, a combination of enzymatic activation (including POD and CAT) and glutathione metabolism was the main pathway for the hybrid Zheyouza 1510. However, in Zheyou 51, the ascorbic acid metabolic pathway was the most important process for the scavenging of the excessive ROS. Therefore, for improvement in rapeseed quality, one should pay attention to the effects of genotypic variation on the elimination of ROS when utilizing rapeseed blanket seedling transplanting technology.

## Figures and Tables

**Figure 1 antioxidants-13-01022-f001:**
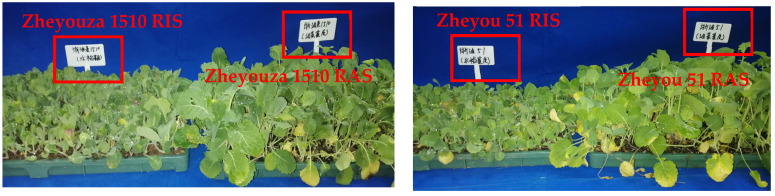
Phenotype of blanket rapeseed seedlings at 30 d old: Zheyouza 1510 and Zheyou 51 varieties under rice growth substrate (RIS) and rapeseed growth substrate (RAS) in the red box.

**Figure 2 antioxidants-13-01022-f002:**
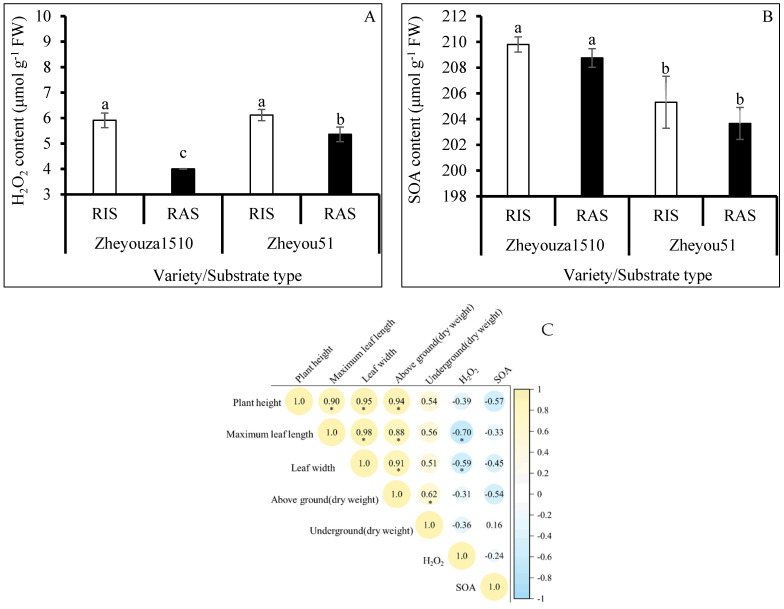
ROS production under rice and rapeseed growth substrates in Zheyouza 1510 and Zheyou 51 and the correlations between ROS contents and seedling quality: (**A**) hydrogen peroxide; (**B**) superoxide anions; and (**C**) Pearson’s correlation coefficients between ROS and seedling quality. Different lowercase letters indicate a significant difference among treatments using Duncan’s method (*p* < 0.05). “*” indicates a significant difference at *p* < 0.05. Error bars indicate the standard deviation (SD) values.

**Figure 3 antioxidants-13-01022-f003:**
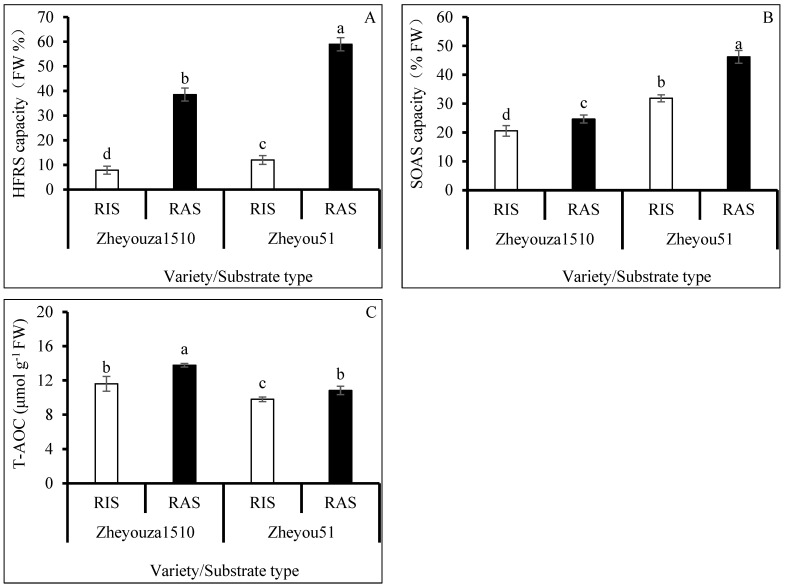
ROS scavenging capacity under rice growth substrate and rapeseed growth substrate in Zheyouza 1510 and Zheyou 51: (**A**) hydroxyl free radical scavenging capacity (HFRS); (**B**) superoxide peroxide anion scavenging capacity (SOAS); and (**C**) total antioxidant capacity (T-AOC). Different lowercase letters indicate significant difference among treatments using Duncan’s method (*p* < 0.05). Error bars indicate standard deviation (SD) values.

**Figure 4 antioxidants-13-01022-f004:**
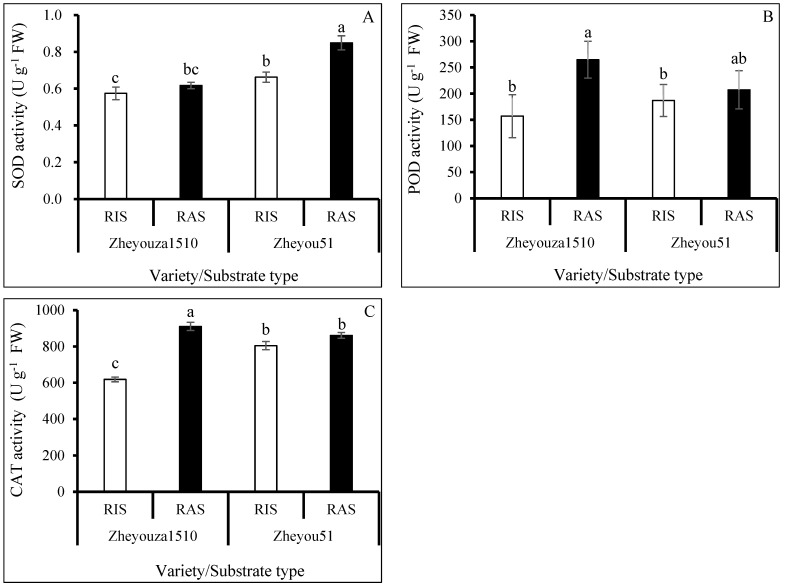
Alteration of enzyme activities in rapeseed leaves under rice growth substrate and rapeseed growth substrate in Zheyouza 1510 and Zheyou 51: (**A**) superoxide dismutase (SOD); (**B**) peroxidase (POD); and (**C**) catalase (CAT). Different lowercase letters indicate significant difference among treatments using Duncan’s method (*p* < 0.05). Error bars indicate standard deviation (SD) values.

**Figure 5 antioxidants-13-01022-f005:**
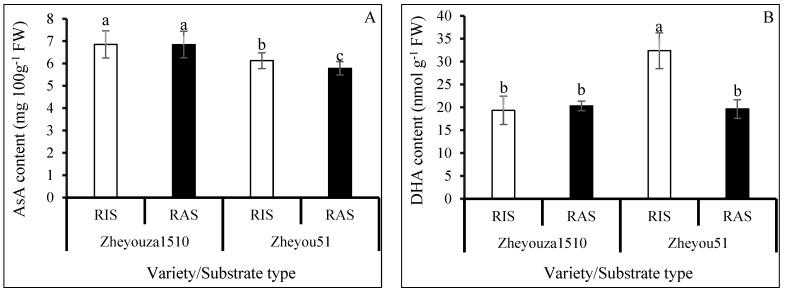
Analysis of ascorbic acid metabolism in rapeseed leaves under rice growth substrate (RIS) and rapeseed growth substrate (RAS) in Zheyouza 1510 and Zheyou 51: (**A**) reduced ascorbic acid (AsA); (**B**) oxidized ascorbic acid (DHA); (**C**) dehydroascorbate reductase (DHAR); (**D**) monodehydroascorbate reductase (MDHAR); (**E**) ascorbic acid oxidase (AAO); (**F**) ascorbate peroxidase (APX); and (**G**) L-galactose-1,4-lactone dehydrogenase (Gal LDH). Different lowercase letters indicate significant difference among treatments using Duncan’s method (*p* < 0.05). Error bars indicate standard deviation (SD) values.

**Figure 6 antioxidants-13-01022-f006:**
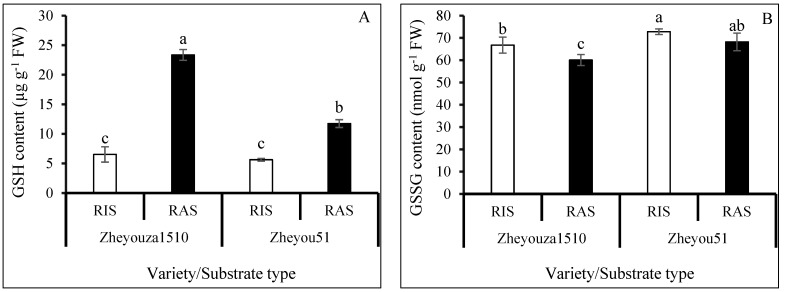
Analysis of glutathione metabolism in rapeseed leaves under rice growth substrate (RIS) and rapeseed growth substrate (RAS) in Zheyouza 1510 and Zheyou 51: (**A**) reduced glutathione (GSH); (**B**) oxidized glutathione (GSSG); (**C**) glutathione peroxidase (GPX); (**D**) glutathione reductase (GR); (**E**) glutathione-S-transferase (GST); (**F**) thioredoxin peroxidase (TPX); (**G**) glutamate cysteine ligase (GCL); and (**H**) total sulfhydryls (TSHs). Different lowercase letters indicate significant difference among treatments using Duncan’s method (*p* < 0.05). Error bars indicate standard deviation (SD) values.

**Figure 7 antioxidants-13-01022-f007:**
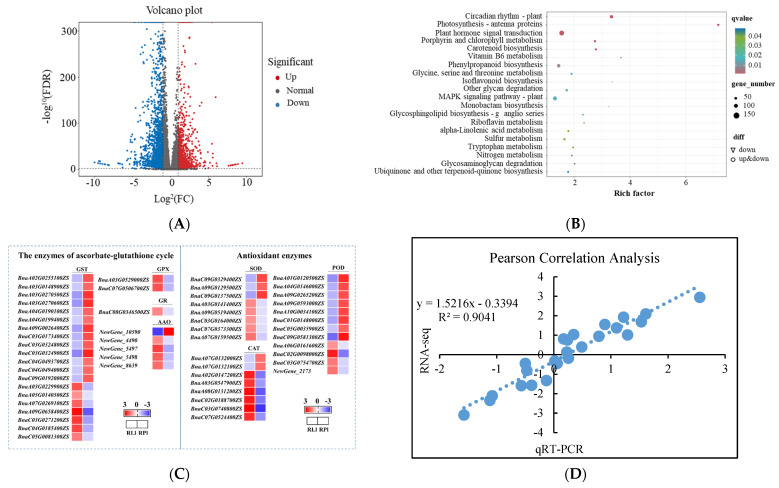
RNA-seq analysis on effects of rice growth substrate (RIS) and rapeseed growth substrate (RAS) on seedling quality in Zheyouza 1510: (**A**) volcano plot of differentially expressed genes under two crop growth substrates; (**B**) KEGG analysis of differentially expressed genes under two crop growth substrates; (**C**) identified genes in enzymatic and non-enzymatic systems for ROS scavenging; and (**D**) correlation analysis between expression levels of selected genes via RNA-seq and qRT-PCR.

**Table 1 antioxidants-13-01022-t001:** Agronomic performance of rapeseed seedlings under rice growth substrate (RIS) and rapeseed growth substrate (RAS) in Zheyouza 1510 and Zheyou 51.

Rapeseed Variety	Substrate	Plant Height (cm)	Maximum Leaf Size (cm)		Dry Matter (g plant^−1^)	
			Length	Width	Aboveground	Underground
Zheyouza 1510	RIS	1.47 ± 0.78 c	3.97 ± 0.24 c	2.97 ± 0.41 c	0.42 ± 0.04 c	0.10 ± 0.01 a
	RAS	3.23 ± 0.19 b	6.15 ± 0.17 b	4.84 ± 0.19 b	0.52 ± 0.02 b	0.11 ± 0.01 a
Zheyou 51	RIS	1.61 ± 0.48 c	3.87 ± 0.43 c	3.16 ± 0.30 c	0.38 ± 0.06 c	0.07 ± 0.01 b
	RAS	5.63 ± 0.07 a	6.70 ± 0.23 a	5.78 ± 0.26 a	0.79 ± 0.04 a	0.13 ± 0.01 a
Cultivar (C)		**	ns	*	**	**
Substrate (S)		**	**	**	*	ns
C × S		ns	ns	ns	ns	ns

Note: different lowercase letters in the same column indicate a significant difference among the treatments using Duncan’s method (*p* < 0.05). “*”, “**”, and ns indicate a significant difference at *p* < 0.05, that at *p* < 0.01, and no significant difference, respectively.

## Data Availability

Data are available upon reasonable request.

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
