# Peer review of "Can Rice Growth Substrate Substitute Rapeseed Growth Substrate in Rapeseed Blanket Seedling Technology? Lesson from Reactive Oxygen Species Production and Scavenging Analysis"

_antioxidants, 2024, doi:10.3390/antiox13081022_

Round 1

Reviewer 1 Report

The reviewed article is very interesting. It addresses the relatively new problem of growing rapeseed on a rice growing medium, which appeared mainly in China. The submitted manuscript contains many interesting results that may influence the method of rapeseed cultivation presented in this work and improve it. However, there are a few things that need to be clarified before it's published (Please see detal coments).

1. Introduction is extensive and discusses the problem presented in the article in a very detailed way. It does not require any corrections.

2. I suggest the authors read the article carefully again to eliminate minor linguistic errors.

3. The weak point of the submitted manuscript is the lack of testing of the substrates used. The presented research clearly shows that they had a significant impact on the growth of rapeseed. Therefore, the substrates should be thoroughly examined. It is known that indicators such as pH, organic carbon content and the availability of macro and microelements are very important in plant cultivation. Maybe it is enough to adjust one of these parameters (e.g. change the pH) to the requirements of rapeseed to eliminate the problem. Since plants react with strong oxidative stress to the rice production substrate, the question should be answered what causes this oxidative stress in plants in this substrate.

4. Planting plants from seedlings is a very old method, but it gives quite good results. The problem is (as also noticed by the authors) the lack of an appropriate number of people who could do this work. The need to employ such workers significantly increases the cost of such cultivation. So, shouldn't we focus on the type of crops that use machines?

Author Response

Reviewer 1

Comments 1: The reviewed article is very interesting. It addresses the relatively new problem of growing rapeseed on a rice growing medium, which appeared mainly in China. The submitted manuscript contains many interesting results that may influence the method of rapeseed cultivation presented in this work and improve it. However, there are a few things that need to be clarified before it's published (Please see detal coments).

Response: Very thank you for your positive comments. We response to your detailed comments below point by point to improve the quality of manuscript. The revised places were high lightened by green background.

Comments 2. Introduction is extensive and discusses the problem presented in the article in a very detailed way. It does not require any corrections.

Response: Thank you for your positive comments.

Comments 3:  I suggest the authors read the article carefully again to eliminate minor linguistic errors.

Response:  Very thank you for your kind suggestions. We used the agency as recommended by the journal to eliminate the linguistic errors in every corner as possible. Please see the certification.

Comments 4: The weak point of the submitted manuscript is the lack of testing of the substrates used. The presented research clearly shows that they had a significant impact on the growth of rapeseed. Therefore, the substrates should be thoroughly examined. It is known that indicators such as pH, organic carbon content and the availability of macro and microelements are very important in plant cultivation. Maybe it is enough to adjust one of these parameters (e.g. change the pH) to the requirements of rapeseed to eliminate the problem. Since plants react with strong oxidative stress to the rice production substrate, the question should be answered what causes this oxidative stress in plants in this substrate.

Response: I totally agree with your point that the substrates should be thoroughly examined. Therefore, we added the data as we made an additional measurement including pH and soil organic matter. You can find the result at L27-128. I also agree with you that the oxidative stress to the rice substrate might be affected by one factor. However, according to the current study, we can find that the greatest possibility is the complicated effects on rapeseed growth because the differences of those factors were detected. Therefore, if we want to answer the question what causes the oxidative stress in plant, further experiments are required. Next stage, we will follow your advices to design experiments such as different pH, different nutrients or their interactions on rapeseed growth and development. Anyway, we thank you for your constructive suggestions.

Comments 5: Planting plants from seedlings is a very old method, but it gives quite good results. The problem is (as also noticed by the authors) the lack of an appropriate number of people who could do this work. The need to employ such workers significantly increases the cost of such cultivation. So, shouldn't we focus on the type of crops that use machines?

Response: I strongly agree with your viewpoint. Actually, blanket seedling transplanting in rapeseed might be applied only in China because of the large area of rice-rapeseed rotation. After rice harvested, direct seeding is too late to reduce rapeseed yield seriously. Although traditional transplanting can improve rapeseed yield, labor-costing is a very big problem. Therefore, machinery transplanting is necessary. In this experiment, we employed hand manual during experimental process which want to catch whether there are problems during application of the technology. In fact, the application of this technology is mechanized. Now I attach the picture below. You are right, we should focus the type of crops for machines, however, as you know, during many agronomic practices, the operators are used to making some minor modifications. The main goal of this modification is to reduce production cost. In China, I can understand their thinkings because the production cost is high as compared to other countries. So, this is the reason we conducted the experiment.

Reviewer 2 Report

This manuscript describes an effort to determine which method of transplantation technology is better (RIS vs RAS) for rape seed production.  Results clearly indicate the RAS is better but there are genetic influences.  The authors want to ascribe these differences to various ROS scavenging systems   While the data seem to support this, I believe the authors go too far in ascribing the differences solely to antioxidant activities.  Certainly, their RNASEQ data point to numerous other possible contributors. The authors should decrease the strength of statements regarding this ( e.g, line 29-30 in abstract). One could change “due to “ to “ is contributed to” 

Also I expect that when RNASEQ data is included in a manuscript that all the data be publicly available. This data must be deposited ina publicly available database (eg., GEO or other database)

All the figures (most have multiple A,B,C panels in each figure) could be consolidated for ease of reading and examining data. 

The description of the methods used (the entire section from line 153-193) is totally insufficient.  If a generally accessible URL which describes the methods in reasonable detail (with citations) is available, it must be included. If not, the scientific basis must be described (with appropriate citations) must be included for each method).  Simply citing a company whose methods are not generally available is not acceptable. 

An additional issue is that the English needs attention (I mention quite a few below, there are too many more to mention). It seems that the introduction and discussion need the most attention in this regard. 

1)    The following section require attention to make the meaning clearer, change word usage or correct English.  Line 11-12, 56-57,67-69, 73-78, 88, 90-92, 465-466, 508-511, 553-554, 571-572, 579, 588-595

2)    Spelling “Person” Line 224, 474

3)    Figure 4, line391, B is repeated 2x with different explanations.

4)    Line 459; Phenylpropanoid and isoflavone are mentioned as enriched; they seem quite minor as they are not nearly so much as other shown (for example, photosynthesis, antenna complexes. Perhaps these other categories and rationale should be briefly mentioned and discussed.

5)    Line 499, 613. Which genes were selected for rtpcr? (Could list in methods section)

6)    Line 520-527, This paragraph seems s out of order, perhaps it c should be introduction to the discussion (i.e., first para in the discussion)?

7)    Line 560-561; I think this an overgeneralization; its only that this pathway does not distinguish between the varieties or treatments. 

8)    Line 132: Is this what was done? If so, it should state so.

Author Response

Reviewer2

Comments 1: This manuscript describes an effort to determine which method of transplantation technology is better (RIS vs RAS) for rape seed production.  Results clearly indicate the RAS is better but there are genetic influences.  The authors want to ascribe these differences to various ROS scavenging systems   While the data seem to support this, I believe the authors go too far in ascribing the differences solely to antioxidant activities.  Certainly, their RNASEQ data point to numerous other possible contributors. The authors should decrease the strength of statements regarding this ( e.g, line 29-30 in abstract). One could change “due to “ to “ is contributed to” 

 Response: I heartfully thank you for your positive comments and constructive suggestion. You are right, we also noticed that the different rapeseed growth performance should be the combination of oxidative stress and others according to our RNAseq DATA. Therefore, we are happy to adopt your suggestion to change those words.

Comments 2: Also I expect that when RNASEQ data is included in a manuscript that all the data be publicly available. This data must be deposited ina publicly available database (eg., GEO or other database)

 Response: Thank you for your suggestion. We have subjected the data to NCBI, which had the reference number, PRJNA1140847. This reference number was also added in the “data” section.

Comments3: All the figures (most have multiple A,B,C panels in each figure) could be consolidated for ease of reading and examining data. 

 Response: Thank you for your suggestion. We re-scaled the figures and consolidated those letters for ease of reading and examining data.

Comments 4: The description of the methods used (the entire section from line 153-193) is totally insufficient.  If a generally accessible URL which describes the methods in reasonable detail (with citations) is available, it must be included. If not, the scientific basis must be described (with appropriate citations) must be included for each method).  Simply citing a company whose methods are not generally available is not acceptable. 

Response: Thank you for your suggestion. Although the methods have detailed instructions with accessible URL, they are in Chinese. Therefore, I decide to add scientific basis with citations.

Comments 5: An additional issue is that the English needs attention (I mention quite a few below, there are too many more to mention). It seems that the introduction and discussion need the most attention in this regard. 

Response: Thank you very much for your suggestion. We have revised the manuscript by agency as recommended by the MDPI journal. The detailed information as you pointed below were also revised.

Comments 5: The following section require attention to make the meaning clearer, change word usage or correct English.  Line 11-12, 56-57,67-69, 73-78, 88, 90-92, 465-466, 508-511, 553-554, 571-572, 579, 588-595

Response: Thank you. We have revised as mentioned.

Comments 6: Spelling “Person” Line 224, 474

Response: Thank you. We have corrected.

Comments 7: Figure 4, line391, B is repeated 2x with different explanations.

Response: Thank you. We have changed.

Comments 8: Line 459; Phenylpropanoid and isoflavone are mentioned as enriched; they seem quite minor as they are not nearly so much as other shown (for example, photosynthesis, antenna complexes. Perhaps these other categories and rationale should be briefly mentioned and discussed.

Response: Thank you very much. We have added those information both in RESULTS and DISCUSSION section.

We have added the content L477~L478and L622~L28. You are right. We also noticed the down regulation of photosynthesis and other physiological changes were measured. We are preparing another manuscript for this regulation.

Comments 9: Line 499, 613. Which genes were selected for rtpcr? (Could list in methods section)

Response: Thank you. We have added information.

Comments 10: Line 520-527, This paragraph seems s out of order, perhaps it c should be introduction to the discussion (i.e., first para in the discussion)?

Response: Thank you very much. We have checked that your suggestion very good and re-adjust their order.

Comments 11: Line 560-561; I think this an overgeneralization; its only that this pathway does not distinguish between the varieties or treatments. 

Response: Thank you for your advice. We have revised this sentence to make the meaning clearly.

Comments 12: Line 132: Is this what was done? If so, it should state so.

Response: Thank you. We have deleted this sentence, which was redundant.

Round 2

Reviewer 1 Report

The article may be published in its current version

The article may be published in its current version

Author Response

Comments 1: The article may be published in its current version.

Response: Thank you very much for your positive comments and kind help.

Reviewer 2 Report

See first review

This manuscript is much improved.  I appreciate the authors inclusion of information about methodology and the appropriate references.

Line 238-9. The only additional change I would suggest is to specify where the RNASEQ data is located (NCBI).

Author Response

Comments: 

This manuscript is much improved.  I appreciate the authors inclusion of information about methodology and the appropriate references.

Line 238-9. The only additional change I would suggest is to specify where the RNASEQ data is located (NCBI).

Response: Thank you very much for your positive comments and kind help. We have added the information as suggested. The English is improved by the angecy as recommended by the joural.